# Cellular Mechanisms Involved in the Combined Toxic Effects of Diesel Exhaust and Metal Oxide Nanoparticles

**DOI:** 10.3390/nano11061437

**Published:** 2021-05-29

**Authors:** Alessandra Zerboni, Rossella Bengalli, Luisa Fiandra, Tiziano Catelani, Paride Mantecca

**Affiliations:** 1POLARIS Research Center, Department of Earth and Environmental Sciences, University of Milano—Bicocca, Piazza della Scienza 1, 20126 Milan, Italy; rossella.bengalli@unimib.it (R.B.); luisa.fiandra@unimib.it (L.F.); paride.mantecca@unimib.it (P.M.); 2Inter-University Center for the Promotion of the 3Rs Principles in Teaching & Research (Centro 3R), 56122 Pisa, Italy; 3Microscopy Facility, University of Milano-Bicocca, Piazza della Scienza 3, 20126 Milano, Italy; tiziano.catelani@unimib.it

**Keywords:** mixtures, metal oxide, nanoparticles, diesel exhaust particles, colony-forming efficiency, autophagy

## Abstract

Diesel exhaust particles (DEPs) and non-exhaust particles from abrasion are two main representative sources of air pollution to which humans are exposed daily, together with emerging nanomaterials, whose emission is increasing considerably. In the present work, we aimed to investigate whether DEPs, metal oxide nanoparticles (MeO-NPs), and their mixtures could affect alveolar cells. The research was focused on whether NPs induced different types of death in cells, and on their effects on cell motility and migration. Autophagy and cell cycles were investigated via cytofluorimetric analyses, through the quantification of the autophagic biomarker LC3B and PI staining, respectively. Cellular ultrastructures were then observed via TEM. Changes in cell motility and migration were assessed via transwell migration assay, and by the cytofluorimetric analysis of E-cadherin expression. A colony-forming efficiency (CFE) assay was performed in order to investigate the interactions between cells inside the colonies, and to see how these interactions change after exposure to the single particles or their mixtures. The results obtained suggest that NPs can either reduce the toxicity of DEPs (CuO) or enhance it (ZnO), through a mechanism that may involve autophagy as cells’ response to stressors and as a consequence of particles’ cellular uptake. Moreover, NPs can induce modification of E-cadherin expression and, consequentially, of colonies’ phenotypes.

## 1. Introduction

The inhalation of airborne ultrafine particles (UFPs, <0.1 μm diameter) is known to promote the insurgence and exacerbation of severe lung diseases, including chronic obstructive pulmonary disease (COPD), lung fibrosis, and lung cancer [1]. Humans have been exposed to airborne UFPs throughout all of our evolutionary stages, but over the last century our exposure to these airborne particles has grown drastically [2], due to the advent of new anthropogenic sources. Several epidemiological studies have suggested that the carcinogenic potential of particulate matter (PM) and UFPs, including diesel-combustion-derived particles, is mainly related to their small size and to the amount of polycyclic aromatic hydrocarbons (PAHs) and toxic metals that are adsorbed on the soot core [3,4]. In urban areas, two main sources of atmospheric pollution are ultrafine diesel exhaust particles (DEPs) and non-exhaust particles from abrasive sources, which are enriched in metal content—especially in Cu, Zn, and Fe—due to tire and brake wear [5,6]. Furthermore, the increasing production of engineered nanoparticles (NPs)—especially metal oxide nanoparticles (MeO-NPs), such as copper and zinc oxide (CuO, ZnO) NPs developed for industrial, medical, and research purposes—is an additional source of emitted particles in the nano-sized range, which could lead to increased human exposure to metal-based toxic compounds [7]. The toxic nature of NPs varies widely depending on their quantity, portals of entry, and duration of exposure. The evaluation and collection of toxicity and exposure data is needed for the individual pollutants. However, humans are inevitably exposed to a mixture of pollutants that could coexist in the same environmental conditions. Despite this evidence, co-exposure to co-contaminants, such as UFPs (e.g., DEPs) and engineered NPs, has largely been ignored so far. Indeed, the interactions between UFPs and NPs, as well as their possible agonist, antagonist, or synergistic effects on biological systems, are still little known. From a perspective of cumulative risk assessment (CRA), it will be important to evaluate the possible interactions between different pollutants, acting on multiple pathways, which could lead to common or different health outcomes. Recently, we presented a study on the combined in vitro effects of DEPs and CuO and ZnO NPs on A549 human lung cells [8]. The data showed that co-exposure to DEPs and NPs may induce physico-chemical interactions between the particles, with consequent increased cytotoxicity for ZnO NPs mixed with DEPs, and reduced effects for CuO NPs when mixed with DEPs. In the present work, we explore some of the mechanisms that could drive the biological outcomes produced by the exposure of lung cells to mixtures of DEPs and CuO/ZnO, and the relevant single particles (DEP, ZnO, and CuO). Several mechanisms of toxicity have been proposed for both NPs and environmental UFPs, including oxidative stress, apoptosis, inflammatory response, DNA damage, cell cycle alteration, autophagy, lysosomal damage, and cell motility alteration [9,10,11,12]. Most of these have been extensively elucidated, while some others have not yet been fully investigated and understood. Regarding NP-induced mechanisms of cell death, several studies reported the activation of the apoptotic pathway by CuO, ZnO, and TiO_2_ NPs, and by PM and ambient UFPs [13,14]. Nevertheless, this is not the only pathway that could be activated by NPs: indeed, high doses could induce irreversible cellular damage and consequent necrosis. Furthermore, there is a strong relationship between necrosis, apoptosis, and even autophagy. Autophagy is a conserved catabolic pathway, stimulated by multiple forms of cellular stress, by which dysfunctional cellular components and aggregated proteins are delivered to the lysosomes for degradation [15]. Recently, autophagy has been recognized as an important response in NP-induced toxicity [16], and can be triggered either by NP-induced oxidative stress or by NP uptake and compartmentalization in autophagolysosomes for degradation, acting as a survival mechanism in response to harmful compounds. However, an excess of autophagy may also lead to cell death. Thus, this mechanism has also been proposed as a key cellular event in the framework of adverse outcome pathways (AOPs) related to lung fibrosis and acute lung toxicity [16]. Another mechanism of lung toxicity induced by UFPs is the dysregulation of cell motility and migration, which could lead to epithelial–mesenchymal transition (EMT). EMT is a biological process that allows a polarized epithelial cell to acquire multiple changes that enable it to assume a mesenchymal cell phenotype, which includes enhanced migratory capacity, invasiveness, and elevated resistance to apoptosis [17]. This process is characterized by the downregulation of cellular epithelial markers (e.g., E-cadherin) and the upregulation of mesenchymal ones (e.g., N-cadherin, Vimentin, etc.) [18]. Some observations indicate that autophagy and EMT are linked in a complex relationship: cells that undergo EMT indeed require autophagy activation in order to survive during metastatic spreading, but on the other hand, autophagy could limit the activation of EMT by acting as an oncosuppressive signal [19].

In this paper, we aimed to investigate whether DEPs, MeO-NPs, and their mixtures could affect human lung adenocarcinoma cell line (A549) functionality and morphology, by assessing cellular apoptosis and necrosis, oxidative stress response, and DNA damage, while also focusing on NP-induced autophagy and its ability to modulate cells’ motility and migration. Autophagy was investigated through flow cytometric quantification of the autophagic biomarker LC3B [20], and the microscopic analysis of LC3B in colocalization with lysosomes. Cells’ ultrastructures were then observed via TEM in order to confirm the presence of autophagosomes in treated cells. Changes in cell motility and migration were assessed via transwell migration assay and through the cytofluorimetric analysis of E-cadherin expression, after the exposure to individual NPs or to mixtures of DEPs and MeO-NPs. Colony-forming efficiency (CFE) assay was performed in our previous study in order to determine NPs’ and mixtures’ cytotoxic and cytostatic effects. However, CFE assay also provides additional information on the cell–cell interactions, intercellular junction, and the ability of cells to aggregate and form colonies. Here, a classification of the colonies, based on their shape and area, was performed. Colonies were also stained for E-cadherin and cytoskeletal actin filaments in order to investigate the interactions between cells inside the colonies. The results obtained suggest that the treatment with mixtures can result in different cellular toxicological effects compared to the individual counterparts, evidencing the importance of including the investigation of combined exposure in future research on the health safety assessment of environmental particulate pollutants.

## 2. Materials and Methods

### 2.1. Preparation of Particle Suspensions and Mixtures 

Commercial CuO (<50 nm, CAS 1317-38-0), ZnO (<50 nm, CAS 1314-13-2), NPs, and diesel exhaust particles (DEPs; NIST SRM^®^2975) were purchased from Sigma-Aldrich (Milan, Italy). Particle suspensions and mixtures were prepared as previously described [8] and characterized (Appendix A). The mixtures were freshly prepared before use. A sub-cytotoxic concentration of DEPs (100 µg/mL) and two concentrations of CuO and ZnO NPs (10 µg/mL and 20 µg/mL, respectively) were used, based on previous results. The particle suspensions were named as follows:

ZnO: suspension of ZnO NPs (final concentration 10 or 20 µg/mL).

ZnO + DEP: mixture suspension of ZnO NPs (10 or 20 µg/mL) and DEPs (100 µg/mL).

CuO: suspension of CuO NPs (final concentration 10 or 20 µg/mL).

CuO + DEP: mixture suspension of CuO NPs (10 or 20 µg/mL) and DEPs (100 µg/mL).

DEP: suspension of DEPs (final concentration 100 µg/mL).

### 2.2. Cell Cultures and Treatments

Human alveolar epithelial cells, A549 (ATCC^®^ CCL-185, American Type Culture Collection, Manassas, VA, USA), were grown and maintained as previously reported [8]. For the experiments, the cells were seeded at a concentration of 1.6 × 10^4^ cells/cm^2^ on 6-well plates (Corning^®^, Pero (MI), Italy) and grown for 24 h. At the optimal confluence (80–90%), the culture medium was replaced with fresh OptiMEM supplemented with 1% foetal bovine serum (FBS) (Gibco, Life Technologies, Monza, Italy), and the cells were treated by directly adding the different particle suspensions (ZnO, ZnO + DEP, CuO, CuO + DEP, or DEP) into the medium. After 24 h of exposure, the cells were processed for the different experimental endpoints. At least three independent experiments were performed for each biological response investigated.

### 2.3. Hoechst 33342/PI Double Staining

Apoptosis and necrosis induced by exposure to mixtures and single particles were assessed by two assays: propidium iodide (PI) and Hoechst 33342 (Hoechst) staining, and annexin V/PI kit.

Hoechst/PI staining allowed us to distinguish viable, apoptotic, necrotic, and mitotic A549 cells after their exposure to NPs and mixtures. At least 300 cells per sample were counted under a fluorescence microscope (Zeiss-Axioplan, Carl Zeiss Microscopy GmbH, Jena, Germany) with a UV filter (365 nm), and scored as viable cells (H-positive and PI-negative cells, without special nuclear characteristics and with an intact plasma membrane), necrotic cells (PI-positive), apoptotic cells (bright H-positive cells with fragmented nuclei), and mitotic cells (H-positive with condensed chromosomes). Data are represented as the mean percentage ± standard error (SE) of the cell for each nuclear characteristic. The description of the annexin V/PI evaluation of apoptotic and necrotic cells by cytofluorimetric analysis is described in Section 2.6.

### 2.4. Cytological Observation

#### 2.4.1. A549 Cell Monolayer Staining

For morphological analysis, the cells were seeded on a cover slide at a concentration of 1.5 × 10^5^ cells/well, cultured for 24 h, and then exposed to particles for a further 24 h. At the end of the treatment, the cells were processed for hematoxylin–eosin (HE) staining, as previously reported [21]. The slides were observed using an optical microscope (Zeiss-Axioplan, Carl Zeiss Microscopy GmbH, Jena, Germany), and pictures were acquired using an AxioCam MRc5 digital camera and processed using AxioVision Real 4.8 software (Carl Zeiss Solutions, Jena, Germany).

For the evaluation of LC3B expression in acidic organelles (lysosomes), the cells were incubated for 2 h with LysoTracker™ Red DND (Thermo Fisher, Monza, Italy), adding 75 nM of probe into the medium. After LysoTracker staining, the cells were fixed in 4% paraformaldehyde for 20 min, and non-specific sites were blocked by incubating the cells with cold PBST (PBS 1X with 0.1% Tween20; Sigma Aldrich) containing 2% bovine albumin serum (BSA). Then, the coverslips were incubated with a rabbit anti-human LC3B primary antibody (1:200 dilution, Cell Signaling, Pero (MI), Italy). Cells were subsequently washed with PBS and incubated with the secondary antibody—goat anti-rabbit Alexa Fluor 488 (1:500, Life Technologies, Monza, Italy)—for 2 h. Finally, the coverslips were dried and observed under an inverted microscope AxioObserver Z1 cell-imaging station (Carl-ZEISS Spa, Milano, Italy). Images were acquired using an MRc5 digital camera and elaborated with ZEN 2.3 Blue edition software (Carl-ZEISS Spa, Milano, Italy).

#### 2.4.2. Colony-Forming Efficiency: Classification of Colonies and Morphological Changes

After the colony-forming efficiency assay, performed as previously reported [8], the cell adhesion and cytoskeletal organization of the colonies was analysed after exposure for 24 h to 10 and 20 µg/mL of NPs or mixtures (100 µg/mL of DEPs and 10–20 µg/mL of NPs). Cells, seeded on petri dishes (Corning^®^, 60 mm diameter), were processed for immunostaining by using rabbit anti-human E-cadherin (1:200 dilution, Cell Signaling) as the primary antibody, and goat anti-rabbit Alexa Fluor 488 (1:500, Life Technologies) as the secondary antibody. In addition, cytoskeletal actin was marked by staining actin microfilaments with rhodamine phalloidin (1:40 dilution, Cytoskeleton Inc., Denver, CO, USA). Nuclei were counter-stained with DAPI (4′,6-diamino-2-phenylindole, 1:100) (Molecular Probes, Life Technologies, Monza, Italia). Colonies were then observed using an AxioObserver Z1 cell-imaging station. Colonies were classified based on four recurrent shapes and morphological features:

Colony type A: A colony composed of a high-density cluster with close interactions between the cells (Appendix A).

Colony type B: A colony characterized by a core of high-density cells and a uniform density distribution of cells all around. This type of colony seems to be composed by the union of two cell clusters (Appendix A).

Colony type C: A colony characterized by a uniform distribution of cells and irregular borders (Appendix A).

Colony type D: A colony composed of dispersed cells with an irregular distribution (Appendix A).

### 2.5. Cell Ultrastructure Analysis

For transmission electron microscopic analysis, the cells were processed after treatment with DEPs (100 µg/mL), NPs (10 µg/mL), and mixtures as described in our previous work [8]. Samples were observed using a Jeol JEM 1220 transmission electron microscope (JEOL, Japan) operating at 80 kV acceleration voltage and equipped with a Lheritier LH72WA digital camera, along with a Zeiss SEM-FEG Gemini 500 operating at 30 kV in STEM mode (Zeiss, Oberkochen, Germany).

### 2.6. Cytofluorimetric Analysis

Cytofluorimetric analyses were performed by using a Cytoflex flow cytometer and the CytExpert software to investigate particles’ and mixtures’ capability of inducing ROS formation, to evaluate the expression of a marker of DNA damage (y-H2AX), to analyse the percentage of live, necrotic, or apoptotic cells (annexin V/PI assay) and the cell cycle progression, and to investigate the expression of the LC3B and E-cadherin proteins in the cells after the treatments.

In order to analyse the apoptosis process after exposure to particles, a dead cell apoptosis kit with annexin V FITC and PI (Thermo Fischer) was used, following the manufacturer’s instructions. Briefly, the cells were seeded and treated with either mixtures or individual NPs, as reported in Section 2.2. After 24 h of exposure, the cells were harvested, centrifuged for 6 min at 1200 rpm, suspended in fresh medium (100 µL), and incubated with annexin V and propidium iodide (PI) for 10 min at RT in the dark, after vortexing. The quantification of the population of apoptotic, late apoptotic, and necrotic cells was analysed through flow cytometry (Cytoflex). Data were expressed as the mean percentage ± SE of the cells for each cell population. At least three independent experiments were performed.

The cell cycle progression of cells exposed to particles and mixtures was investigated after 24 h of exposure by DNA staining. Cells were trypsinized, collected, and pooled with the harvested medium, centrifuged at 1200 rpm for 6 min, fixed in 90% cold ethanol, and stored at −20 °C until analysis. For the analysis, cells were centrifuged at 1600 rpm for 6 min in order to discharge ethanol; samples were resuspended in PBS supplemented with RNase, DNase-free (1 mg/mL, Sigma-Aldrich, Italy) and incubated for 30 min. Finally, the fluorescent dye PI was added to stain the DNA of cells for 7 min in the dark. Fluorescence was measured using a flow cytometer (CytoFLEX, Beckman Coulter GmbH, Krefeld, Germany) with 617 nm bandpass filters. For the analysis, cells in different cell cycle phases (subG0; G1; S; G2/M) were selected and analysed as the mean percentage of cells in each phase.

For ROS detection, A549 cells were pre-incubated for 20 min with the probe carboxy-2′,7′-dichlorofluorescein diacetate (carboxy-DCFDA, 5 μM, Life Technologies) and incubated with NPs for 180 min. After incubation, the cells were detached, centrifuged at 1200 rpm for 6 min, resuspended in 500 μL of PBS, and analysed in the FITC channel using flow cytometry (CytoFLEX) and the CytoExpert software. The signal from the unloaded samples (cells without the probe carboxy-DCFDA) was evaluated as a reference by which to assess cells’ and NPs’ autofluorescence. These values were then subtracted from the values of DCFDA-stained samples. The capability of particles and mixtures to induce the expression of phosphohistone H2AX (y-H2AX), as a marker of DNA double-strand breaks (DSBs), was evaluated by probing exposed and control cells with the rabbit mAb anti-yH2AX Alexa Fluor 488 conjugate (Cell Signalling). After 24 h of exposure to particles, the cells were detached, washed in PBS, fixed in 1% paraformaldehyde in PBS, suspended in 90% cold methanol, and stored overnight at −20 °C before analysis. Samples were then stained for y-H2AX following manufacturers’ instructions and analysed using the CytoFLEX (Beckman Coulter, Milan, Italy) in the FITC channel.

Furthermore, the expression of specific proteins was investigated via cytofluorimetry. Cells, after exposure to 10 µg/mL of NPs, 100 µg/mL of DEPs, and mixtures of particles for 24 h, were harvested, fixed with 4% paraformaldehyde for 15 min, permeabilized with methanol 90%, and then stained with the rabbit anti-human LC3B antibody (1:400 dilution, Cell Signalling) and E-Cadherin (1:200 dilution, Cell Signalling). After 1 h of incubation in the appropriate buffer (0.5% bovine serum albumin in PBS 1X), the cells were washed in buffer and then incubated for 60 min with the secondary antibody AlexaFluor anti-rabbit 488 (1:200 dilution, Life Technologies). After washing, cells were resuspended in PBS and analysed using the CytoFLEX in the FITC channel.

### 2.7. Cell Migration Assay

Cell migration after the exposure to individual NPs and to mixtures was assessed using a cell migration assay kit (Cell Biolab CytoSelects ^TM^). Briefly, 2.2 × 10^4^ cells were seeded on the apical side of an insert (Ø 8 µm) in cell-free medium, treated with particles (DEP 100 µg/mL, NPs 10 µg/mL and mixtures), and incubated for 24 h at 5% CO_2_, 37 °C. To the basal side of the inserts we added 1.5 mL of OptiMEM with 10% FBS. After the incubation time, non-invasive cells on the apical side of the insert were removed with a cotton-tipped swab. Then, the inserts were transferred to a well containing a cell-staining solution and incubated for 10 min. The inserts were then transferred to an empty well, adding an extraction solution, and incubated for 10 min. The extraction solution was then transferred to a 96-well plate, and the OD was measured using a multiplate reader (Infinite 200Pro, TECAN, Milan, Italy) at a wavelength of 560 nm.

### 2.8. Statistical Analysis

The data represent the mean ± standard error of the mean (SE) of at least three independent biological experiments. Statistical analyses were performed using GraphPad Prism software, using the unpaired *t*-test, one-way or two-way ANOVA, and post-hoc test.

## 3. Results

### 3.1. Effects on Cell Viability and Cell Cycle Alteration after Exposure to Mixtures and Single NPs

Cell viability after exposure to single NPs, DEPs, and mixtures was previously assessed via MTT assay [8]. The previous data showed that CuO NPs induce high mortality, while co-exposure with DEPs reduced the toxicity of CuO. DEP + ZnO was slightly more toxic than ZnO NPs alone. However, co-exposure enhanced the toxicity to cells compared with samples treated only with DEPs. The differences observed were attributed to the amount of Cu^2+^ and Zn^2+^ ions released from NPs, either alone or in co-presence with DEPs. Here, we investigated the mechanisms of cell death induced by the different treatments. The present study began with an investigation of the type of death induced by particles. Two different approaches were adopted: a microscopic analysis with Hoechst/PI staining, and cytofluorimetric analysis of annexin V/PI expression. A concentration of 20 μg/mL was previously chosen, since it was the most effective concentration; however, in order to investigate the mechanisms of action of the particles and mixtures, a lower concentration (10 μg/mL) was used for the further experiments. Data from Hoechst/PI staining confirmed the data from the MTT test, showing that CuO NPs at high concentrations (20 μg/mL) induced necrosis. An increase of necrotic cells, though not statistically significant, was observed also after exposure to ZnO, DEP + ZnO, and DEP + CuO (Appendix A). The Hoechst/PI method allows us to classify cells as apoptotic, mitotic, or viable based on their nuclear morphology, and as necrotic cells based on the incorporation of PI inside the cells.

In parallel, annexin V/PI double-staining was performed on cells treated for 24 h with NPs at concentrations of 10 (Figure 1A) and 20 µg/mL (Figure 1B), either in mixture with DEPs or as individual NPs. To detect cell death, an annexin V/PI double-staining kit was used in flow cytofluorimetric analyses. The annexin V corresponding signal is a sensitive method for detecting cellular apoptosis, while propidium iodide (PI) is used to detect necrotic or late apoptotic cells, characterized by the loss of the integrity of the plasma and nuclear membranes. Data generated by flow cytometry are shown in Figure 1, and expressed as the percentage of cells of the different statuses (viable, apoptotic, necrotic, or late apoptotic). In our previous work, concentrations of 25 μg/mL of CuO and ZnO induced strong cytotoxic effects after 24 h of exposure, which reduced the number of viable cells considerably. At a concentration of 10 μg/mL, CuO NPs increased programmed cell death (apoptosis), and a significant reduction in the number of viable cells was observed. At 20 μg/mL, CuO NPs caused an increase in necrotic cells, both alone and in mixture with DEPs. This assay did not show an increase in the number of dead cells after exposure to ZnO and DEP + ZnO, which was expected in keeping with previous results from the viability test (MTT). Moreover, Hoechst/PI staining (Appendix A) confirmed the significant induction of necrosis in cells exposed to CuO (20 μg/mL), with similar results for DEP + CuO, albeit to a lower extent. Hoechst/PI staining also showed an induction of necrosis after ZnO (20 μg/mL), and of both necrosis and apoptosis after DEP + ZnO exposure.

Cell death has been described by a different set of both morphological and biochemical features, which collectively define this process (e.g., loss of cell volume or cell shrinkage and nuclear condensation) [22]. Cell morphology and particle interactions were investigated via optic microscopy on A459 cells exposed to NPs (20 μg/mL), DEPs (100 μg/mL), and mixtures. The HE staining showed that diesel particles interact with cells and cause a morphological alteration of cells. (Figure 1(Cf)). Exposed cells showed a slight increase in elongated cell morphologies with all treatments compared to control samples. The cytotoxic effects of CuO itself induced cell shrinkage and massive cell death, as shown in Figure 1(Cd).

The effects on cell cycle progression are presented in Figure 2. A marked reduction in G1 phase cell percentage was evident after exposure to CuO (10 μg/mL), but also after exposure to DEP + CuO. A corresponding increase in S and/or G2 phase cells was noticed, indicating a possible accumulation of genotoxic effects. The ZnO NPs and their mixtures with DEPs induced slight variations in cell cycle progression compared to the controls, with a slight increase of the percentage of S phase cells with ZnO (10 µg/mL) and DEP + ZnO (10 µg/mL).

### 3.2. Effects on Cell Adhesion and Migration

Here we evaluated the migration of single cells after exposure to NPs (10 μg/mL), DEPs, and mixtures (Figure 3A), as well as the expression of E-cadherin (Figure 3B), using cytofluorimetric analysis. A slight, though not significant, increase in migration was recognizable after all of the treatments, except for CuO, compared to the control samples. Presumably, the high cytotoxicity induced by CuO NP exposure limited the number of migratory cells able to cross the transwell porous membrane. A downregulation of E-cadherin was evident after exposure of cells to ZnO, DEP + ZnO, and CuO, suggesting that these tested substances interfere with the cellular contacts.

In addition, we proposed a method for A549 colony classification utilizing images taken with stereo and fluorescence microscopy directly from cell culture dishes. On the basis of CFE assay of untreated cells, four different types of colonies (Figure 4) have been identified, and this classification could be used as a parameter to evaluate the capability of DEPs, NPs, and mixtures to affect the colonies’ phenotypes. The different types of colonies were identified based on shape, cell distribution, and cell–cell interactions. A deeper evaluation of colonies’ morphologies was conducted through an immunofluorescence investigation (Figure 4A) (Appendix A). The cytoskeleton and cell adhesion, as well the E-cadherin expression, appeared regular in colony types B and C, while in colony type A there was an overlap of cells, which limited the characterization of cytoskeleton and cell–cell adhesion. In colony type D cells presented altered cytoskeletons and reduced expression of E-cadherin.

In Figure 4C are reported the percentages of the four different types of colonies after exposure to particles for 24 h. The analyses were conducted on three independent experiments and reported as the percentage of colonies belonging to each of the four classes with respect to the total number of colonies evaluated. Colony type D, which has a more disorganized morphology and more dispersed cells, was frequent after exposure to all of the treatments, with a percentage around 20%, compared to the 10% present in control cells. This type of colony was prevalent after exposure to CuO (32% with 20 μg/mL and 38% with 10 μg/mL). On the other hand, the percentage of colony types B and C, which presented high cell density, decreased after exposure to CuO NPs. A slight reduction in the prevalence of colony type C was also evident after exposure to DEP + ZnO and DEP + CuO mixtures. CuO at high concentration (20 μg/mL) also induced an increase in the prevalence of colony type A, which are colonies with a small diameter and cells that are very adherent to one another. This result fits with the strongest evidenced cytostatic effect observed for CuO NPs at 20 µg/mL in our previous work. In addition, the area (μm^2^) of each colony type after treatment with particles was evaluated based on three independent experiments (Appendix A), and these analyses confirm the reduction of the colonies’ area after exposure to CuO and its mixtures; a significant decrease in the prevalence of colony type D area was especially evident.

### 3.3. Induction of Autophagy

The induction of the autophagic mechanism by NPs, either alone or in mixture with DEPs, was here evaluated by cytofluorimetric and microscopic analyses of the LC3B biomarker, and by TEM observation of cells that had internalized the tested particles.

Data from cytofluorimetric analysis show that, overall, MeO-NPs, either alone or in co-exposure with DEPs, increased the levels of LC3B expression in exposed cells (Figure 5A), while DEPs alone did not induce an increase in LC3B expression. A significant increase of the marker’s expression was observed with ZnO + DEP, CuO, and CuO + DEP. Since autophagosome clearance depends on its fusion with lysosomes, the colocalization of LC3B puncta and lysosomes has been also performed via immunofluorescence microscopy. The mixture of DEP + ZnO, as well as CuO, induced an increase in LC3B signalling, and the LC3B puncta colocalized with the lysosomes. The ultrastructural analysis of cells exposed to particles shows that CuO NPs increased the intracellular formation of multilamellar bodies (MLBs) (Figure 6E), as well as ZnO and DEPs. In the Figure 6E, an endosome containing particles is also evident, localized between the two MLBs indicated by blue arrows. Autophagosomes, indicated by yellow arrows, are present in cells exposed to DEP + ZnO (Figure 6C) and DEP + CuO (Figure 6F).

## 4. Discussion

In real environmental conditions it is quite difficult to distinguish background and incidental exposure to nano-sized airborne particles, since the methods employed generally measure the presence of UFPs, and do not differentiate between the various types of particles that may be present. Chemical risk assessment has usually focused on single stressors, which also concerns air pollution, because the intense air pollution problems of the 20th century were caused by a limited number of chemicals, which were emitted in considerable quantities. Moreover, in the last years, aside from common air pollution sources (e.g., diesel exhaust particles and biomass burning), new emergent sources have been introduced, such as nanomaterials. It must be considered that the exposure to a single chemical/contaminant is an exception rather than the rule; indeed, in everyday life, humans are exposed to a multitude of different stressors [23]. Today, available data on the joint toxicity of environmental contaminants are very limited. In this perspective, we analysed how the in vitro effects of two widely used metal oxide nanomaterials (CuO and ZnO NPs) can change in concomitance with the presence of DEPs. In a previous study from our group, the pivotal role of released ions in the NPs’ toxic outcomes, and how this could change with the co-presence of DEPs, has been demonstrated [8]. We observed that Cu ions released from the surface of partially soluble CuO NPs were limited by a passivation effect of DEPs, which can explain the lower toxicity of the DEP + CuO mixture. As to ZnO NPs, they provoked cytotoxicity through the release of zinc ions, which seemed to be almost unaffected by the presence of DEPs. The slightly enhanced toxicity observed after lung cells’ exposure to DEP + ZnO mixtures was correlated with an increased NP uptake by the cells when in co-presence with DEPs. Here, the analysis of the mechanisms of cell death involved showed once again that CuO NPs are the most cytotoxic particles, and that they induce apoptosis or necrosis, according to the concentrations tested (10 μg/mL and 20 μg/mL, respectively). Lai et al. reported that the expression of the apoptosis inhibitor proteins Bcl-2 and Mcl-1 significantly decreased in A549 and BEAS-2B cells treated for 24 h with 10 μg/mL of CuO NPs. These observations are in line with our results, confirming that low concentrations of CuO induce apoptosis (Figure 1). The same authors also assumed a correlation between ROS generation and the induction of autophagy after exposure to CuO NPs [24]. At the tested conditions, we were not able to detect any intracellular ROS generation, and only DEPs were able to induce a slight, though not significant, increase in ROS (Appendix A). However, the induction of ROS, and the consequent triggered toxicity by MeO-NPs is controversial, and it is possible that a different ROS detection method could give different results. MeO-NPs’ biological effects could be either ROS-dependent or -independent [10], and in the presence of NPs, the autophagic pathway could be activated by the compartmentalization of harmful compounds in lysosomes for degradation, acting as a survival mechanism [25]. DNA damage from oxidative stress, here evaluated through the analysis of y-H2AX expression, was not observed, in agreement with the lack of ROS generation in our experimental conditions (Appendix A). However, data from cell cycle analysis indicated that exposure to NPs significantly affected the normal progression of cells through the different cycle phases, with an evident arrest of cells in the S and G2/M phases when exposed to either CuO or DEP + CuO, or to a lesser extent, to ZnO and ZnO + DEP. This evidence supports the hypothesis that NPs can alter DNA, consequently leading to programmed cell death by preventing cells from continuing their cycle when DNA defects are detected at the specific checkpoints. The cell cycle can be indeed blocked or delayed in reaction to different genotoxic stresses, but also to structural dysfunctions of proteins [26]. Furthermore, our studies were focused on important mechanisms involved in the response to particle exposure and in the regulation of cancer progression: (1) cell motility and cell-to-cell interaction, and (2) autophagy [27]. Cell motility and the interactions between cells have a role in the maintenance of the homeostasis of the alveolar epithelium functionality in physiological conditions. Alteration of cell motility could also be due to downregulated expression of epithelial markers, such as E-cadherin, which could lead to epithelial–mesenchymal transition (EMT) [28]. This process has been shown to be crucial in tumorigenesis, in addition to imparting a migratory and invasive capacity to cancer cells. The involvement of EMT as a biological response to DEP exposure has been reported before by other authors; testing the toxicity of DEPs to bronchial epithelial cells in vitro, a varying degree of expression of EMT markers was found [29,30]. In our study, however, DEPs did not induce a significant downregulation of E-cadherin, nor did they increase cell migration. Moreover, the morphology of A549 colonies exposed to DEPs did not show peculiar variations with respect to control colonies, suggesting that in this experimental condition the cell–cell adhesion, by which cells interact and attach to neighbouring cells, was not compromised, nor was the process of colony formation. CuO NP exposure, on the other hand, induced a significant decrease in E-cadherin expression and deeply changed the colonies’ phenotypes, with an increased presence of colonies with less adherent cells, here identified as colony type D (Figure 4). This evidence indicates that CuO NPs strongly reduce the adhesiveness of cells inside the colony, and this phenomenon could be the result of an augmented motility and invasiveness capability of cells exposed to such particles. Reduced E-cadherin expression was observed even in cells exposed to ZnO NPs and DEP + ZnO mixtures, but not in those exposed to DEP + CuO. This evidence supports our previous results, in which the biological effects exercised by CuO NPs were limited by the simultaneous presence of DEPs, while DEP + ZnO mixtures may lead to higher toxic outcomes compared to the individual ZnO particles.

The choice to investigate carcinogenic-related responses, such as cell motility and a marker of EMT in a tumoural cell line (A549), could be considered quite controversial. However, this cell line more likely reflects the target of the tested particles, and in vitro studies about EMT and autophagy have been previously performed successfully on these cells [31,32]. However, longer times of exposure and different concentrations of NPs should be tested in order to investigate the carcinogenic process in this cell line in a more relevant manner, but this is beyond the scope of the present work. To obtain more reliable results regarding the interactions between NPs and DEPs, a different range of NP concentrations should also be tested, since different effects could occur that were not recorded at the concentrations used here. The autophagy pathway has been studied before in relation to exposure to DEPs and engineered NPs [33,34]. DEPs emitted from Euro 4 and Euro 5 engines were proven to induce a significant increase of the autophagic flux in bronchial epithelial cells [35]. Moreover, some NPs, including metal-based ones, have been identified as a novel class of autophagy activator [36,37]. NPs can be commonly observed within the autophagosomes, suggesting that the activation of the autophagic pathway is prompted by the effort to sequester and destroy materials that enter into the cytoplasm [38]. Microtubule-associated protein 1A/1B light chain 3B (LC3B) is the most widely used marker of autophagosomes [39], and it was used here as biomarker of autophagy. Our results suggest that CuO NPs lead to the expression of some features connected to the autophagic pathway, in particular the increased signalling of the LC3B marker, and the presence of LC3 puncta colocalized with lysosomes. Furthermore, exposure to DEP + ZnO mixtures also induced the expression of the LC3B marker of autophagy in treated cells. Moreover, exposure to both DEP + ZnO (Figure 6C) and DEP + CuO also caused the formation of autophagosomes. However, it is interesting to note that exposure to ZnO, DEPs, and CuO caused an increase in MLBs, which play a secretory function in human lung alveolar cells. The lysosomal composition of both the autophagic vacuole and the MLBs underlines a relationship in the biogenesis of the two organelles, as previously demonstrated [40], suggesting that an induction in the autophagy pathway may also cause an increase in MLBs. The activation of the autophagy process is likely due to the higher uptake of particles (CuO NPs and DEP + ZnO) in exposed A549 cells with respect to other NPs and mixtures (as reported in [8]), rather than being due to oxidative stress, since in the tested conditions no ROS were observed. Moreover, data from lysosomes and LC3B puncta colocalization strongly suggest that CuO NPs and DEP + ZnO mixtures, once internalized in lung cells, are sequestered in autophagolysosomes for their degradation, as a mechanism of survival against harmful compounds. Our results are consistent with the study conducted by Sun et al. [41], which showed that CuO NPs induce autophagic cell death in A549 cells. Evidence of the involvement of autophagy after exposure to ZnO NPs was reported in a recent study by Liu et al., in which autophagy was identified as the main mechanism of cell death involved in response to ZnO NPs [42]. Altogether, this evidence once again confirms the less toxic potential of CuO NPs when in mixture with DEPs, and the slightly higher toxic potential of DEP + ZnO compared to ZnO NPs alone.

## 5. Conclusions

The assessment and mitigation of the effects of air pollutants is still a challenge for environmental policy, and this topic should be addressed with a more complex approach, based on the evaluation of the combined effects of cumulative exposure to different particulate sources. Furthermore, sophisticated sublethal endpoints linked to chronic toxicity should be considered. In the present work, a simplified model of cumulative exposure was simulated by exposing lung cells to individual NPs or a mixture of MeO-NPs and DEPs. We confirmed that exposure to the NP mixtures can result in different cellular toxicity than their NP counterparts alone. NPs can either reduce the toxicity of DEPs (CuO) or enhance it (ZnO) through a mechanism that involves autophagy as a consequence of cellular uptake of particles. These results support the growing evidence that autophagy is a relevant molecular event in response to UFPs and NPs, and that it should be investigated in future research for the definition of AOPs related to inhalation exposure, since it is also relevant in the combined effects of combustion-derived UFPs and NMs. A modification to the cell–cell interaction of epithelial cells in response to NPs was also observed. A new method for A549 colony identification and classification was proposed, based on the evaluation of their size and shape, and by analysing cellular interactions inside the colonies through E-cadherin expression. This method allows us to understand how cell–cell interaction could be altered by exposure to particles, giving insights about the contribution of particles’ interactions with lung cells to EMT and lung cancer progression. CuO particles, both individually and in mixtures with DEPs—as well as ZnO NPs and, to a lesser extent, their mixtures with DEPs—modified the expression of adhesion molecules and, consequentially, the colonies’ formation, morphology, and dimensions. Further investigations are needed in order to elucidate whether other parameters—such as dose and time of exposure, type of NPs, and sub-chronic experiments—can influence these results. Considering the limitations posed by the use of tumour cell lines, such as A549, in the future, advanced in vitro models (e.g., lung organoids and 3D reconstructed tissues, also derived from human primary cells) should be employed in order to test the effects of metal oxide NPs in mixture with combustion particles, especially for genotoxic and carcinogenic endpoints. Moreover, the use of reference materials should not provide a complete scenario of the cumulative effects of particulate matter with other pollutants. Indeed, the use of particles directly sampled from emissions sources should be considered to be a better choice for testing the cumulative effects of different airborne pollutants. However, the present research—for the first time, to our knowledge—highlights how some relevant mechanisms already known to be involved in response to environmental particles could be differently activated when co-exposure to different particulate sources occurs.

## Figures and Tables

**Figure 1 nanomaterials-11-01437-f001:**
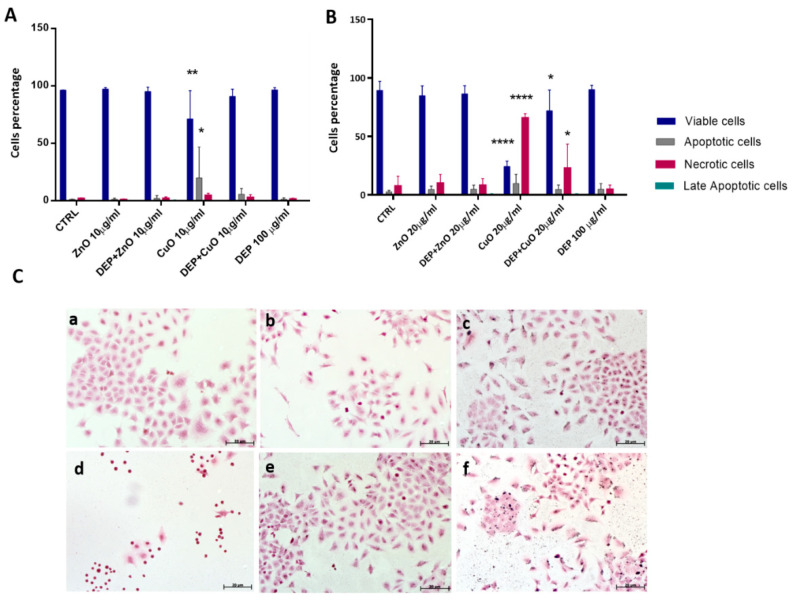
Cell death after exposure to particles. Panels A and B show flow cytometry analyses of cells stained with annexin V and PI after exposure to ZnO and CuO NPs at (**A**) 10 μg/mL and (**B**) 20 μg/mL, either alone or in mixture with DEPs (100 μg/mL). Statistically significant with respect to the control according to one-way ANOVA, ** *p* < 0.01, * *p* < 0.05, and **** *p* < 0.0001 vs. control. (**C**) Morphological analysis. Microscope images of A549 cells, stained with hematoxylin/eosin: (**a**) control; (**b**) ZnO (20 μg/mL); (**c**) DEP + ZnO (20 μg/mL); (**d**) CuO (20 μg/mL); (**e**) DEP + CuO (20 μg/mL); (**f**) DEPs (100 μg/mL); scale bar = 20 μm.

**Figure 2 nanomaterials-11-01437-f002:**
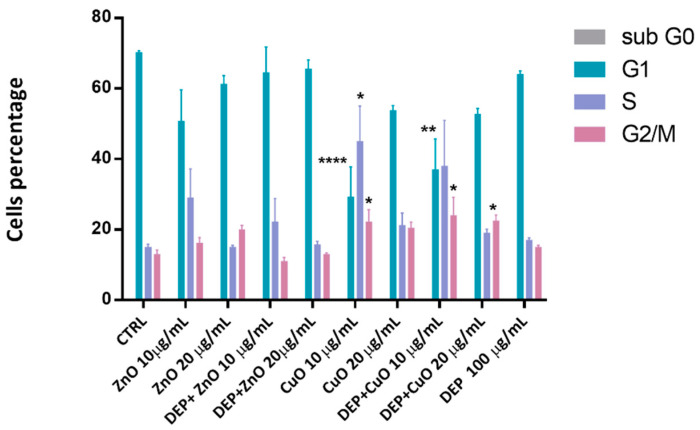
Cell cycle investigation. Cell percentage in each phase of the cycle after 24 h exposure to particles and mixtures (ZnO and CuO at 10 and 20 μg/mL). Statistical analysis was performed using one-way ANOVA with Dunnett’s multiple comparisons tests. ** *p* < 0.01, * *p* < 0.05 and **** *p* < 0.0001 vs. control.

**Figure 3 nanomaterials-11-01437-f003:**
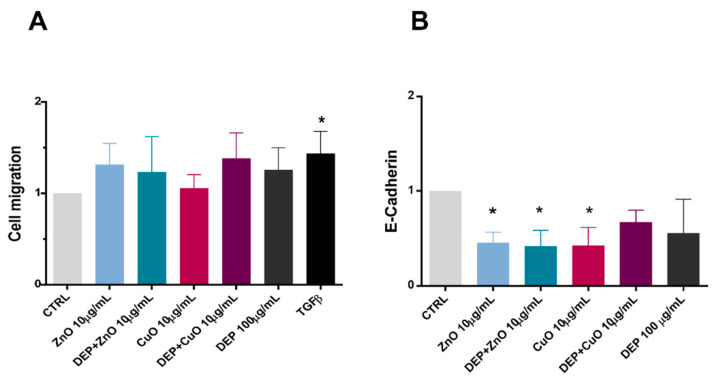
Cell migration and E-cadherin expression. (**A**) Cell migration was evaluated in cells exposed to 10 μg/mL of ZnO and CuO, and respective mixtures with DEPs (100 μg/mL), compared with untreated cells (CTRL) and with a positive control for migration (TGF-β) (data are presented as fold change with respect to the control). (**B**) At the same exposure concentration, the expression of E-cadherin (presented as fold change with respect to the control) was evaluated by cytofluorimeter. Data represent the mean ± SE of three independent experiments (*n* = 3). *: Statistically different with respect to control samples; *p* < 0.05, one-way ANOVA and Bonferroni’s post-hoc test.

**Figure 4 nanomaterials-11-01437-f004:**
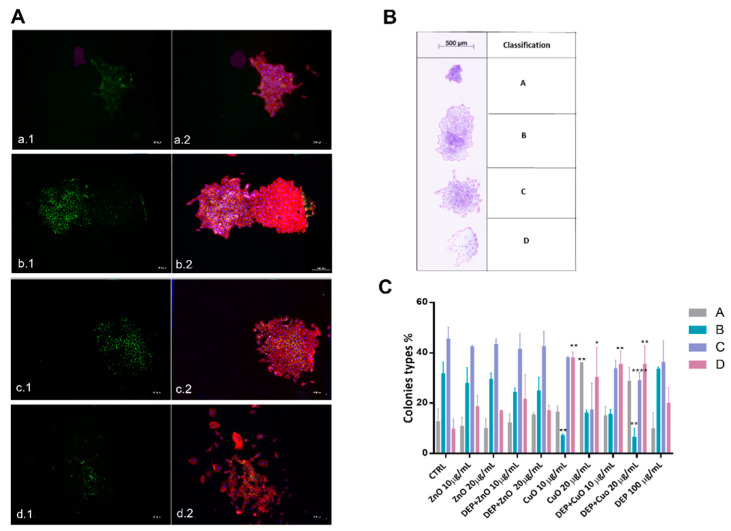
Colonies’ classification and the percentage prevalence of each colony type after cells’ exposure to particles. Panel (**A**) shows fluorescence microscope images of colonies stained for E-cadherin (Green) (a.1; b.1; c.1; d.1), rhodamine phalloidin (Red), and DAPI (Blue) (a.2; b.2; c.2; d.2). Colonies’ classification according to their shape and cellular contacts is shown in panel (**B**). Histograms representing the percentage of colonies in the four groups (A, B, C, and D) are represented in panel (**C**). Statistical analysis was performed using one-way ANOVA with Dunnett’s multiple comparisons tests. ** *p* < 0.01, * *p* < 0.05, and **** *p* < 0.0001 vs. control.

**Figure 5 nanomaterials-11-01437-f005:**
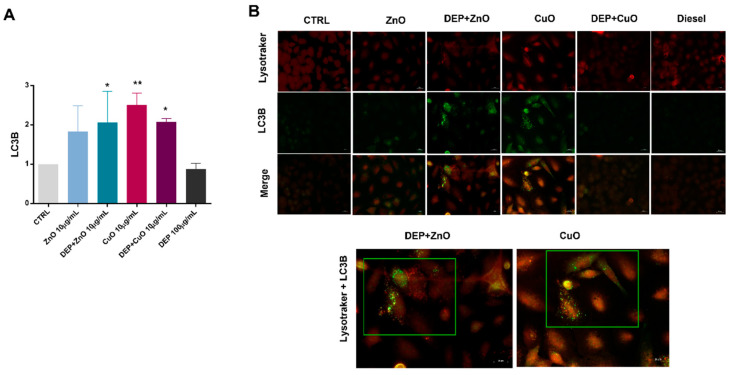
Autophagy detection by LC3B and colocalization with lysosomes. (**A**) The expression of LC3B, as a marker of autophagy, was evaluated by cytofluorimetric analysis in A549 cells exposed for 24 h to ZnO and CuO NPs (10 µg/mL), alone or in mixture with DEPs (100 µg/mL), and histograms represent the mean ± SE of the fold change of three independent experiments (*n* = 3). In panel (**B**) are shown fluorescence microscopy images of A549 cells exposed to either MeO-NPs alone or mixtures, and stained for lysosomes (Lysotracker, red) and LC3B (green). Merged figures indicate the colocalization of LC3B puncta with acidic organelles (highlighted images of cells exposed to DEP + ZnO and CuO NPs). Statistical analysis was performed using one-way ANOVA with Dunnett’s multiple comparisons tests. ** *p* < 0.01 and * *p* < 0.05.

**Figure 6 nanomaterials-11-01437-f006:**
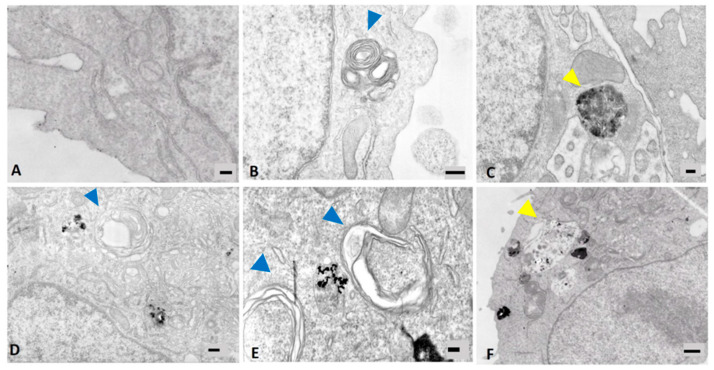
TEM analyses of A549 cells after 24 h exposure to particles. (**A**): ctrl (100 nm); (**B**): ZnO 10 μg/mL (300 nm); (**C**): DEP + ZnO 10 μg/mL (200 nm); (**D**): DEP 100 μg/mL (300 nm); (**E**): CuO 10 μg/mL (200 nm); (**F**): DEP + CuO 10 μg/mL (200 nm). Blue arrows indicate multilamellar bodies (MLBs); yellow arrows indicate autophagosomes.

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
