# Peer review of "Cellular Mechanisms Involved in the Combined Toxic Effects of Diesel Exhaust and Metal Oxide Nanoparticles"

_nanomaterials, 2021, doi:10.3390/nano11061437_

Round 1
Reviewer 1 Report
The paper may be published in its current form
Author Response
We thank the reviewer.
Reviewer 2 Report
The manuscript represents significant contribution to the simultaneous effects of disel exhaust particles and metal oxides nanoparticles in the environment. The results reported here further developed data presented by the same team of authors in 2019 (reference 8) where the same model and the same suspensions of particles were used, but different endpoints were measured and discussed. Paper is well written, introduction summarizes current knowledge, methods are adequate, results are well described and statistical power is convincing. Discussion and conclusions reflect the data, but some study limitations should be included.
Major comments:
- A549 is human lung tumor cell line, which is widely used in nanotoxicology as model of epithelial lung cells. Authors repeatedly described in the text (for example line 101 and elsewhere) A549 as type II human lung epithelial cells, which is not correct. A549 is tumor cell line used as model...Its karyotype is very different from type II cells...This should be corrected in the text.
- As mentioned above, some limitations of the study should be mentioned in Conclusions. First of all, real human exposures might be by far different from tho model used in this study: as mentined above, tumor cell line A549 is different from type II cells, SRM of DEP and commercially available particles of ZnO and CuO do not cover complex mixtures of particulate environmental pollutants resulting from transportation...
Minor points:
- line 635: ..."toxic potential"...instead of ..."toxicological potential"...
- line 567: ..."and ZnO +DEP. This evidence"...instead of "and ZnO+DEP. this evidence"...
Author Response
Major comments:
- We correct in the text the description of A549 cells as type II human lung epithelial cells. A549 are human lung adenocarcinoma cells but this cell line was established has properties of Type II cells of the pulmonary epithelium(Foster et al. 1998)(Lieber et al. 1976).
- We agree with the reviewer and we added in the conclusion a consideration on the limits of this work. The use of tumor cell line (A549) could be considered as a limitation of the work, a better evaluation of Metal Oxide in mixture with combustion particles should be investigated in more complex in vitro models (e.g., lung organoids). Moreover, the use of Standard materials should not provide a complete scenario of the cumulative effects of particulate matter with other pollutants (Line 665…).
Foster, Kimberly A. et al. 1998. “Characterization of the A549 Cell Line as a Type II Pulmonary Epithelial Cell Model for Drug Metabolism.” Experimental Cell Research 243(2): 359–66.
Lieber, Michael et al. 1976. “A Continuous Tumor‐cell Line from a Human Lung Carcinoma with Properties of Type II Alveolar Epithelial Cells.” International Journal of Cancer 17(1): 62–70. https://pubmed.ncbi.nlm.nih.gov/175022/ (May 25, 2021).
Minor points:
- We corrected the mistake in the text.
- We corrected the mistake in the text.
Reviewer 3 Report
The authors present a solid study that demonstrates the modifying effect of metal oxide nanoparticles on diesel exhaust particles. Different functional and morphological approaches were used for the experimental investigation, which were adequately described and apparently correctly performed. The results presented suggest that metal oxide nanoparticles can modify the cell toxic effect of diesel exhaust particles and, depending on the case, attenuate or enhance it.
The presentation is good and the study could stimulate further systematic research in this area. It is true that no systematic study is presented here that systematically covers all aspects over large areas, but that is not to be expected here.
In summary, I recommend the publication of the manuscript presented. Only in the discussion and conclusion do I recommend a more cautious formulation, especially since only a small concentration range of metal oxide nanoparticles is covered. biphasic effects could occur that were not recorded at the concentrations used.
Author Response
We thank the reviewer; we added a phrase at Line 604 regarding the limitation of the concentration tested in the present study.